# Visual Measurement System for Wheel–Rail Lateral Position Evaluation

**DOI:** 10.3390/s21041297

**Published:** 2021-02-11

**Authors:** Viktor Skrickij, Eldar Šabanovič, Dachuan Shi, Stefano Ricci, Luca Rizzetto, Gintautas Bureika

**Affiliations:** 1Transport and Logistics Competence Centre, Transport Engineering Faculty, Vilnius Gediminas Technical University, Saulėtekio al. 11, 10223 Vilnius, Lithuania; viktor.skrickij@vilniustech.lt (V.S.); gintautas.bureika@vilniustech.lt (G.B.); 2Institute of Land and Sea Transport Systems, Technical University of Berlin, Strasse des 17. Juni 135, 10623 Berlin, Germany; dachuan.shi@tu-berlin.de; 3Dipartimento di Ingegneria Civile, Edile e Ambientale (DICEA), Sapienza Università di Roma, Via Eudossiana 18, 00184 Roma, Italy; stefano.ricci@uniroma1.it (S.R.); luca.rizzetto@uniroma1.it (L.R.)

**Keywords:** railway, track gauge measurement, track geometry, lateral displacement of the wheelset, image processing, in-service trains

## Abstract

Railway infrastructure must meet safety requirements concerning its construction and operation. Track geometry monitoring is one of the most important activities in maintaining the steady technical conditions of rail infrastructure. Commonly, it is performed using complex measurement equipment installed on track-recording coaches. Existing low-cost inertial sensor-based measurement systems provide reliable measurements of track geometry in vertical directions. However, solutions are needed for track geometry parameter measurement in the lateral direction. In this research, the authors developed a visual measurement system for track gauge evaluation. It involves the detection of measurement points and the visual measurement of the distance between them. The accuracy of the visual measurement system was evaluated in the laboratory and showed promising results. The initial field test was performed in the Vilnius railway station yard, driving at low velocity on the straight track section. The results show that the image point selection method developed for selecting the wheel and rail points to measure distance is stable enough for TG measurement. Recommendations for the further improvement of the developed system are presented.

## 1. Introduction

The transport sector is a vital sector of the economy in European Union countries. However, there are many challenges, such as congestion in urban areas, oil dependency, emissions, and unevenly developed infrastructure, that need to be solved. In tackling these challenges, the railway sector needs to take on a larger share of transport demand in the next few decades.

For both passenger and freight railway transport, safety is of great importance. Rolling stock and railway infrastructure must meet safety requirements concerning their construction and operation. A part of the safety requirements refers to the rules for the technical conditions of railway assets during exploitation. As the assets are subject to degradation, they are regularly inspected and maintained to ensure that their conditions are within the safety limits [1]. For instance, track geometry (TG) maintenance is one of the most important activities for railway infrastructure. The track is inspected by specific track recording coaches (TRC) at set time intervals. More frequent inspection can generate more information on TG degradation over time, leading to more reasonable maintenance decisions. This finally results in a higher level of safety and reliability for railway infrastructure. However, in practice, the frequency of inspection is mainly limited by the availability of TRC and track, considering that the inspection should not affect the regular traffic. Therefore, it is desirable to conduct condition monitoring for TG on in-service vehicles. This research work aimed to develop a wheel–rail lateral position monitoring method that would improve the inertial system used for TG.

This investigation was aimed at testing the accuracy of the visual measurement and proving the possibility of its usage for track gauge evaluation. The achieved results and contribution to the field of science are as follows: (i) the accuracy of the visual measurement of the distance between image points using calibrated test samples is high enough to implement it for TG measurement tasks; (ii) the field test results show that the image point selection method developed for selecting the wheel and rail points for measuring distance is stable enough for TG measurement; (iii) recommendations for the further improvement of the developed system are presented. This proves the adaptability of a low-cost visual measurement system for wheel–rail lateral position evaluation and provides further research directions for improving its performance.

This paper is organised as follows: In Section 2, the latest TG monitoring techniques are reviewed; the gaps are identified. In Section 3, methods developed for wheel–rail lateral position monitoring employing stereo imaging are presented. In Section 4, the results of the experimental investigation are presented. In Section 5, the discussion is presented, and further steps for technology development are presented.

## 2. Related Work

In 2019, the TG measurement system market was valued at USD 2.8 billion and was proliferating; it is likely to reach USD 3.7 billion by 2024 [2]. Major system manufacturers are AVANTE International Technology (Princeton Junction, NJ, USA), DMA (Turin, Italy), Balfour Beatty (London, UK), R. Bance & Co. (Long Ditton, UK), MER MEC (Monopoli, Italy), Fugro (Leidschendam, Netherlands), MRX Technologies/Siemens Mobility Solutions (Munich, Germany), KLD Labs (Hauppauge, NY, USA), ENSCO (Springfield, VA, USA), TVEMA (Moscow, Russia), Plasser & Theurer (Vienna, Austria), Graw/Goldschmidt Group (Gliwice, Poland/Leipzig, Germany), and others. These companies have three types of solutions: (i) for train control systems (TCS), (ii) for in-service vehicles, and (iii) manual measurement systems. The solutions used on TCS are expensive and require continuous maintenance. The measurement periodicity using a track geometry car (TGC) is low (from one month to twice per year) due to the high operational costs and track possession [3]. TG measurement using in-service vehicles is not widespread yet; however, this may allow the continuous monitoring of track conditions in a cost-efficient way [4]. The usage of in-service vehicles for TG measurement in urban areas brings many advantages as well [5]. The performance of manual measurement systems is low; they may be used for specific tasks.

To ensure the safety of railway operations, TG parameters, including the track gauge, longitudinal level, cross-level, alignment, and twist, are usually inspected using TRC. The measurement requirements are provided in standard EN13848-1:2019 [6]. Many researchers have analysed the TG measurement task. Chen [7] proposed the integration of an inertial navigation system with geodetic surveying apparatus to set up a modular TG measuring trolley system. Kampczyk [8] analysed and evaluated the turnout geometry conditions, thereby presenting the irregularities that cause turnout deformations. Madejski proposed an autonomous TG diagnostics system in 2004 [9]. Analysis of the railway track’s safe and reliable operation from the perspective of TG quality was analysed by Ižvolta and Šmalo in 2015 [10]. Different statistical methods can be used to evaluate the TG condition and make a prediction model for its degradation. Vale and Lurdes presented a stochastic model for the prognosis of a TG’s degradation process in 2013 [11]. Markov and Bayesian models were used for the same task by Bai et al. and Andrade and Teixeira in 2015 [12,13]. Chiachío et al. proposed a paradigm shift to the rail by forecasting the future condition instead of using empirical modelling approaches based on historical data, in 2019 [14].

Different methods to estimate TG parameters from vehicle dynamics and direct measurements have been proposed. The first type of system is an inertia-based compact and lightweight no-contact measuring system that allows an accurate evaluation of railway TG in various operational conditions. The acceleration sensor may be placed on the axle box, bogie frame, or car body. Bogie displacement can be estimated by the double integration and filtering of the measured accelerations. It is an indirect measurement; such a system has some drawbacks. For example, it requires the consideration of vehicle weight, which may change in a wide range of in-service vehicles, and the acceleration depends on it. Additionally, the absolute value of acceleration may exceed almost 100 *g,* which occurs due to short-wavelength railway track components and, simultaneously, accelerations caused by long-wavelength irregularities due to the wheelset oscillation phenomena often being below 1 *g*, which makes the estimating of long-wavelength components challenging and requires high-accuracy accelerometers that support a wide-enough range of measured accelerations [4]. The method works reasonably well in the vertical direction. The results of the lateral measurements made with different vehicles are so widely scattered that it is virtually impossible to draw any conclusions [15]. The most challenging is the estimation of lateral TG due to the relative wheel–rail motion in the lateral direction, which occurs on account of the peculiar geometry of wheel–rail contact [4]. During measurements of the track gauge, the flanges of both wheelset wheels are not in contact at the same time. A few cases are possible: (i) the left wheel flange is contacting the left rail of the track; (ii) the right wheel flange is contacting another rail of the track; (iii) the wheels’ flanges are not in contact with any rail of the track (the “centred” position of the wheelset on the track). This means that the wheel’s displacement is not directly representative of the track’s alignment as happens with the estimation of the vertical irregularity components. The lateral alignment is hard to measure accurately without optical sensors using only an inertial system [16]. To solve this issue, the displacement achieved from the lateral acceleration at the vehicle and lateral wheel–rail position is needed.

The second type is multiple-sensor systems that can be mounted on a suitable vehicle to implement TG measurement. Using the latest noncontact optical technology that provides direct measurement, such as laser-based or light detection and ranging (LIDAR)-based, accurate results can be achieved. An example of a multiple-sensor system is a mobile track inspection system—the Rail Infrastructure aLignment Acquisition system (RILA), developed by Fugro. This system uses several sensors combining both direct and indirect measurements [3]: a global navigation satellite system (GNSS); an inertial measurement unit (IMU); LIDAR; two laser vision systems; three video cameras. It can be mounted on in-service vehicles, and the axle load issue is also solved; however, the price of such a system is high due to the number of sensors used.

The third type is the model-based methods proposed by Alfi and Bruni in 2008, and Rosa et al. in 2019 [4,17]. The acceleration measured at different positions (the axle boxes, bogie frame, or car body) is used to identify the system’s inputs. This can be done using a pseudoinversion of the frequency response function matrix derived from a vehicle mathematical model; however, lateral measurement uncertainty remains.

Different sensors were reviewed, searching for new techniques for TG measurement in the lateral direction. There is a list of requirements for such sensors: particular dimensions and weight, energy consumption, lifecycle costs, system robustness, measurement accuracy, technical compatibility, output data format, and measurement repeatability. The importance of these requirements is not equal. In 2017, Xiong et al. used a three-dimensional laser profiling system, which fused the outputs of a laser scanner, odometer, IMU, and global positioning system (GPS) for rail surface measurement [18]. The authors used such a system to detect surface defects such as abrasions, scratches, and peeling. Laser-based systems have high measurement accuracy and a high sampling rate. However, the price of the system was not suitable for broad application on in-service trains. In 2011, Burstow et al. used thermal images to investigate wheel–rail contact [19]; the authors found that such technology can be used to monitor the condition of assets such as switches and crossings and understand the behaviour of wheelsets through them. In 2019, Yamamoto extended the idea of Burstow et al. [19] and proposed using a thermographic camera for tests for rail climb derailment [20]. It is possible to extend such a system for gauge measurement. However, the resolution of the thermographic camera is low. Therefore, the geometry calculated from the pixels will have a relatively high error. Visual detection systems (VDSs) are used for detecting various defects in railways.

Marino and Stella used a VDS for infrastructure inspection in real time in 2017 [21]. The proposed system autonomously detected the bolts that secure the rail to the sleepers and monitored the rail condition at high speeds, up to 460 km/h. VDSs are widely used for rail surface defect detection [22,23,24,25]. There are studies where the detection of defects of the railway plugs and fasteners using VDSs is analysed [26,27]. In 2019, Singh et al. developed a vision-based system for rail gauge measurement from a drone [28]. Such a method allows measuring track gauges with errors ranging from 34 to 105 mm; the solution is more appropriate as an additional source of information for inspection. Karakose et al. used a camera for the fault diagnosis of the rail track, as well, in 2017 [29]. The camera was placed on the front of the train. Following this procedure, the distance between the rails was calculated from the pixels. The camera was placed relatively far away from the track; therefore, the pixel count per measured distance decreased and affected the accuracy negatively. To maximize the track gauge measurement accuracy, the camera should be placed as close as possible to the measured area and have as narrow a field of view (FOV) as possible while including the full measurement area. A solution can be the use of a separate camera for each rail, and using the known distance between the cameras for estimation.

Summarising, current TG measurement systems are complex and combine different types of sensors; thus, their high cost precludes wide deployment in in-service trains. Inertia-based noncontact measuring systems allow an accurate evaluation of railway TG in a vertical direction, but the lateral measurement results are not reliable enough. A new robust and low-cost solution for lateral wheel–rail position measurement is needed.

## 3. Methods

In Section 3, an algorithm developed for wheel–rail lateral position evaluation based on visual measurements is presented. It consists of two steps. The first is the estimation of the distance between two selected points in stereo images. Experiments were carried out to determine the visual measurement method’s accuracy by application for object width estimation. The second is the detection of measurement points for wheel–rail lateral position estimation in a sequence of images. It was developed and tested on real sequences of wheel–rail contact images, recorded during field tests. The test platforms, the algorithms, and their testing are described in this section.

### 3.1. An Algorithm for Object Width Estimation Using a Stereo Camera

A platform for the real-time testing of sensing solutions in the laboratory was created (Figure 1). It included a Stereo Labs ZED stereo camera (1) connected to a power-efficient artificial intelligence embedded system, an NVIDIA Jetson TX2 (2); a notebook computer (3), used to control the NVIDIA Jetson TX2; a 220/12V inverter (4); and a 12V battery (5). There is not enough natural lighting under the rail vehicle, and additional lighting is required all the time. Therefore, the experiments were carried out using two types of artificial light sources: a cold cathode fluorescent lamp (CCFL) in the laboratory, and a light-emitting diode (LED) for the field tests. During the laboratory test, 2 halogen lights of 3000 lm each with parabolic reflectors were used at a distance of about 3 m from the measured object. During the field tests, two LED lights of 1700 lm each were used with each stereo camera at a distance of about 1000 mm from the measurement areas. The setup with LED lights was portable enough, and the measurement data could be stored internally in the NVIDIA Jetson TX2 or transmitted to remote data storage via Wi-Fi or ethernet.

The algorithm for the estimation of an object’s width described here can be used with the stereo depth and visual image information that is captured using a stereo camera. The object’s width in millimetres was estimated from the point count between its vertical edges in the image. The measured object was selected by the operator. The trigonometrical equations presented in Appendix A were used to calculate the actual object width in millimetres from the distance in image points. The algorithm is presented in Figure 2a, and the scheme for the measurement is presented in Figure 2b. The image was captured using a calibrated stereo camera and displayed on the computer monitor (Figure 1a). An example of an image is presented in Figure 3a.

The user selects the object, whose width is measured with a blue cross using the computer mouse cursor. The distance from the camera to the object under the cursor is displayed next to the cursor if the depth image is available (Figure 3a). After a point is selected, a Canny edge detector is used to obtain the image edges, including the edges of the object (Figure 3b). After the edge detection, the vertical edges that are the closest to the selected measurement point are identified using the iterative algorithm, which scans the image horizontally on both sides of the selected point. All the tests were performed with the assumption that the measured surface of an object is parallel to the image sensor plane to simulate the lateral displacements of the object with respect to the camera.

For the tests, the distance between the camera and the object was determined using stereo vision, validated using a Leica laser distance meter with an error of ±1.5 mm, and varied in the range of 750 to 1250 mm. This range of distances was selected based on the limitations of the stereo camera and the visual width measurement accuracy’s correlation with distance. The closest distance was selected based on minimal focus and a stereo depth measurement distance of 750 mm, while the farthest distance was selected to be at 1250 mm, so the measured object would appear as big as possible in the camera’s FOV. Additionally, this distance range appears to be right for installing a camera on the train bogie. It is important to add that the camera was installed above the surface on which the specimens were placed (Figure 1a) and tilted down so the farthest object would be near the centre of the FOV (Figure 3a). This setting closely reassembled the camera position with respect to the measured area during the field test. The absolute similarity was not crucial, as our width measurement algorithm accounts for the object’s offset from the centre of the FOV.

Precision specimens (Figure 4) for width measurement were used during the experiments. The widths of the specimens used in the experiment were 5.12 mm ± 0.2 µm; 10.24 mm ± 0.3 µm; 15.36 mm ± 0.3 µm; 21.5 mm ± 0.3 µm; 25 mm ± 0.3 µm; 35.24 mm ± 0.4 µm; 40.36 mm ± 0.4 µm; 46.5 mm ± 0.4 µm; 50 mm ± 0.4 µm; 55.12 mm ± 0.5 µm; and 60.24 mm ± 0.5 µm. The specimens had a shiny surface that complicated the measurements. However, the same conditions are expected during real-world measurements, as both the running surfaces of wheels and rails can have shiny surfaces.

The developed algorithm uses integer coordinates. Based on Equations (A1)–(A8), presented in Appendix A, and the measurement scheme presented in Figure 2b, Equation (1) for the theoretical measurement accuracy in millimetres is derived:(1)$$A=d_{C2\mathrm{OX}}\cdot 2\sin\left(\frac{a_{\mathrm{PIX}}}{2}\right)=d_{C2\mathrm{OX}}\cdot 2\sin\left(\frac{a_{HFOV}/w_{I}}{2}\right)$$
where $d_{C2\mathrm{OX}}$ is the distance to the object in millimetres; $a_{\mathrm{PIX}}$, the angle per point in degrees; $a_{HFOV}$, the horizontal FOV angle in degrees; and $w_{I}$, the FOV width in points.

Based on Equation (1), to increase the measurement accuracy, the distance between the camera and object should be minimal. The camera’s FOV should be selected to fill the frame as much as possible at the selected distance.

The results of the width measurement are presented in Section 4.1. The manual selection of the object and image output was used for visual presentation and testing purposes. The measurement point selection is further automatised by the second algorithm.

### 3.2. Algorithm for Wheel–Rail Lateral Position Evaluation Method Using Stereo Imaging

The algorithm presented in Section 3.1 is needed to determine the accuracy of the lateral position evaluation. The accuracy and repeatability requirements are presented in standard EN13848-1:2019 [6]. The measurement system was developed and installed in a TRC Plasser & Theurer EM-140 (Figure 5a). It is used in Lithuanian Railways and has all the equipment needed for the TG measurement and validation of the developed measurement system. The research intended to focus on developing and testing measurement equipment that allows the detection of the wheelset–track position to extend inertial TG measurement systems’ functionality. The system measures the track gauge and, after combining it with measurements by the inertial system, allows the estimation of the lateral alignment.

The Plasser & Theurer EM-140 is adapted for railways with a 1520 mm track gauge. The developed measurement system combined two stereo cameras, one for each side of the wheelset, with two LED light sources. A frame for camera fixation was developed (Figure 5b); the cameras were connected to the frames using dampers designed for testing purposes. Four artificial light sources were installed on the frame as well. The whole measurement system was fixed on the Plasser & Theurer EM-140 bogie frame. A schematic view of the whole setup and camera position is presented in Figure 6.

An algorithm for wheel–rail lateral position evaluation was developed based on the visual distance measurement technique described earlier (Figure 7). A solution for the automatic selection of measurement points instead of manual selection was developed. This was needed for evaluating the distance between the wheel and rail, taking into account the complex geometry and its variation due to wear processes.

The wheel–rail lateral position measurement algorithm (Figure 7) involves the estimation of the coordinates of two measurement points; see Figure 8. The coordinates of the first measurement point (1) on the wheel flange are estimated by applying the calibrated shift in pixels from the detected wheel flange point (2) that is being tracked in a sequence of images. The second measurement point (3) on the rail edge from the wheel flange side is estimated based on the parameters of the detected rail edge line (4) and vertical coordinate of the wheel flange measurement point (1). The vertical coordinate of the second measurement point is set equal to the vertical coordinate of the wheel flange point, and the horizontal coordinate is evaluated by solving linear equation (2) for the horizontal coordinate of the rail edge line at the vertical coordinate of the wheel flange point (1). After that, the distance is evaluated between the selected points using the algorithm described earlier (Figure 2a).

Some assumptions can be made, simplifying detection and allowing substantial processing speed improvements for wheel flange point detection and tracking. Analysing the video data, it was found that the displacement of the wheel was significantly smaller compared to the displacement of the rail because the camera was mounted on the bogie frame and its displacement was limited with respect to the wheel. The camera’s vibration is possible while passing the rail joints and in the case of lateral wheel to rail contact. However, the camera alignment returns to the initial position after excitation. The wheel flange edge geometry does not change during the measurement, and template matching can be used.

Experimentally, it was found that template convolution with an image provides point-accurate location even if small changes in the edge-detected image occur. Other methods such as greyscale template matching, feature matching and homography were also tested, but they were too sensitive to lighting changes or too flexible, and random errors appeared. To track the exact point location in the image, only a small part of an image that is at least twice the template’s size is processed. This is possible because the movement of the wheel relative to camera is limited as discussed earlier. The rectangular shape region of interest is centred on the last detected location of the template, allowing speeding up the convolution. To improve the measurement stability, the detected point coordinates are filtered using a low pass filter. Vertical wheel displacement can be almost ignored; only the wheel-to-rail contact position is essential, as it determines the point where the rail measurement point is on a vertical axis. The horizontal wheel movement data are filtered to track the wheel position with a minimal delay compared to the filter for vertical movement, but still reducing the measurement noise and vibrations.

The rail edge measurement is based on a line-tracking algorithm provided in the open-source Visual Servoing Platform library (ViSP) by Inria Lagadic [30]. The algorithm detects and tracks points on selected lines and the direction of the edge and approximates the edge line based on the detected edge positions for each point. This algorithm allows defining the range of detection, threshold and point sample step. We used the smallest sample step of 1, thresholds between 5000 and 50,000 and a detection range of 20 points. After the edge line is detected, the formula (2) is used for calculating the horizontal location of the rail measurement point based on the coordinates of the measurement point on the rail edge $x_{R}$:(2)$$x_{R}=(y_{L}-y_{W})/\tan\left(\theta \right)+x_{L}$$
where $x_{L}$, $y_{L}$ are the coordinates of one point of a detected line; $y_{W}$ is the vertical coordinate of the measurement point on the wheel flange; and θ is the inclination angle of the detected rail edge line.

During processing, the results for the wheel–rail distance and track gauge are calculated and saved to a text file. Additionally, for debugging purposes or real-time visualisation, an overlay on the video can be shown. Figure 8 shows an example of such a visualisation (frames from recorded system output). As the recording of all visual data is not needed, the required storage space is reduced a lot. This is the main advantage of real-time visual data processing on an NVIDIA Jetson TX2 or AGX. Moreover, a program can be modified to send results to the centralised data acquisition unit. This type of computing is currently called edge computing, as the source visual data are not transmitted and are processed in place.

## 4. Results

In Section 4, the results achieved during the laboratory and field tests are presented. The first investigation that was conducted in the laboratory was used to determine the accuracy of the visual distance estimation method. The second investigation was used for testing the wheel–rail lateral position evaluation method; it was performed using Lithuanian Railway facilities.

### 4.1. Algorithm for Wheel–Rail Lateral Position Evaluation Method Using Stereo Imaging

Using Equation (1), the theoretical measurement accuracy of the developed algorithm was calculated. For images captured with a ZED stereo camera, with a horizontal FOV $a_{HFOV}$ = 74.37^0^ and selected horizontal resolution $w_{I}$ = 1920 points, for the distance between the object and the camera $d_{C2\mathrm{OX}}$ = 1000 mm, the theoretically possible measurement accuracy is $A_{\mathrm{ZED}}$ = 0.676 ≈ 0.7 mm. Such values are acceptable, as for gauge measurement, they are defined as being in the interval of ±1 mm in EN13848-1:2019 [6]. The theoretical uncertainty depends on the resolution of the image sensor. Meanwhile, the experimental uncertainty depends on the optical and focus quality. Experiments were carried out, and the results for the measurement errors are presented in Table 1.

During the experiment, the measured objects were illuminated only by artificial light (two CCFLs with a total luminous flux of 6000 lm) for all the 13 specimens (Table 1); the measurements were repeated 50 times. The standard deviation was used to evaluate the precision of the measurement uncertainty. The normality of the distribution of the experimental data was investigated by applying skewness and kurtosis, according to the calculation method presented by Sivilevičius et al. in 2017 [31]. All the measured values were within the measuring range $\bar{X}\pm 3\sigma$, where $\bar{X}$ is the average value and σ is the standard deviation. The normality hypothesis of empirical data was accepted, as the values of skewness and kurtosis satisfied the requirements. The normality of the distribution of the sample sizes was tested, which enabled the calculation of the average precision of the measurements. It can be seen that the measurement error for all the specimens in the range of 5.12 to 65.36 mm was in the acceptable range of ±1 mm. Moreover, for all the 650 measurements (samples multiplied by measurements per sample), the maximal error was in the range of ±1 mm. The visual measurement accuracy for the distance between the image points using calibrated test samples is high enough to implement it for TG measurement.

### 4.2. Algorithm for Wheel–Rail Lateral Position Evaluation Method Using Stereo Imaging

After the investigation conducted in the laboratory, the field test was performed, as described in Section 3.2. The field test was conducted on a railway with a 1520 mm track gauge, which is typical for Lithuanian Railways. The experiment was conducted on a sunny day. The initial tests were performed in the Vilnius railway station yard; the straight track section was selected for investigation, and the train velocity was about 10 km/h. It was an initial investigation to test hardware components such as the lighting, damper of the camera and power supply, and software components such as the data acquisition and data processing algorithms. It was necessary to define further steps to improve the system’s accuracy and reliability. The developed system was installed on the geometry-measuring machine the Plasser & Theurer EM-140 (Figure 5a,b and Figure 6); it allowed us to compare the data we measured with a reference. The reference system measures the gauge but not the wheel–rail lateral position directly, so we tuned our system to measure the gauge as well. The data from the Plasser & Theurer EM-140 were recorded every 25 cm, and they were velocity independent; the sample rate of our image system was 30 frames per second (FPS). In Figure 9, the measurement results for the conventional measurement system installed on the Plasser & Theurer EM-140 and the developed camera-based system are presented. To compare, the developed system measurements were resampled to be comparable to the measurements from the reference system, and the root mean square error (RMSE) between them was calculated, which was equal to 2.81 mm.

As can be seen in Figure 9, the data received from the developed system surpassed and lagged the reference system measurements at different distances; this occurred due to different train velocities and increased the RMSE value. The initial test’s main output is that the developed algorithm stably detects the measuring points on the wheel and the rail.

## 5. Discussion

The investigation was intended to focus on developing and testing a low-cost system that allows the detection of the wheelset–track lateral position to extend the functionality of existing inertial TG measurement systems. To find a suitable solution, it is necessary to understand the specific characteristics of the measured object, which is a wheelset running on a track during rail vehicle operation. Due to the wheelset lateral oscillation phenomenon, the wheelset rotates about axis Y, moves linearly, and rotates about the track longitudinal axis X and vertical axis Z, as well on the tangent, resulting in a sinusoidal motion on the track. The wheelset movement trajectory becomes even more complex and difficult to describe when the rail vehicle runs on track curves and along track superelevation (inclined track).

The achieved results show that the proposed visual wheel–rail lateral position measurement system principle is suitable for track gauge measurement. During the investigation, it was found that it is possible to measure the geometrical parameters with an accuracy that meets the standard EN13848-1:2019 requirements in laboratory facilities [6]; the error was less than ±1 mm for all the measurements.

It was found that the majority of the known algorithms used for geometry evaluation from an image are not suitable for wheel–rail lateral position measurement. The original algorithm was developed; it was validated on the image data obtained in the field measurements. The image processing methods used in the proposed algorithm require finetuning parameter settings, such as parameters for the filters and the line detector. The measurement results rely on the camera specifications, illumination conditions, camera fixation, and train parameters.

During the field test, a ZED stereo camera was used. The camera was oriented so that it could film both the rail edge and wheel flange surface. The camera’s horizontal FOV of 74.37°, covered by 1920 horizontal points, resulted in a ±0.338 mm theoretical accuracy and ±1 mm accuracy based on the presented laboratory test results. At a distance of 1000 mm, not only was the wheel–rail contact filmed but so were a lot of additional objects; they were redundant information for this task. For data processing, only part of the picture was used. It is better to use a camera with the smallest FOV; it increases accuracy. It was found that the relative displacement of the camera installed on the vehicle bogie was relatively small, and the distance between the camera and wheelset could be considered constant. There is no need to use a stereo camera in this case. If the device will be installed on the vehicle body, a stereo camera is the best solution. The IMU sensor inside the ZED camera showed that the acceleration in the vertical direction did not exceed 20 m/s^2^ during the field test. The designed damping system worked satisfactorily; with an increase in train velocity, the vibrations will increase, and this system should be revised.

The rail vehicle mass (loaded/unloaded) should be considered, as it is desired to conduct monitoring for TG on in-service vehicles. As the masses of in-service trains may vary, they may lead to different deformations of the wheelset, and measurement error may appear. Mass measurement/estimation is essential for the TG measurement system as a whole. Before integrating the proposed system into a TG system, the axle load needs to be defined. Our proposal is estimating the axle load considering the longitudinal vehicle dynamics, implementing a recursive least-squares algorithm.

The field tests showed that sunlight orientation might reduce the measurement quality; therefore, an artificial light source is required. During the experiment, two LED alternating-current (AC) light sources for each camera were used with a total luminous flux of 3400 lm. When filming, AC lighting introduces flickering that is seen when using exposure times less than the period of the AC. To decrease this effect, the exposure of the camera can be increased, but this leads to motion blur that decreases the video quality and does not allow using high-framerate cameras taking more than 60 FPS. A constant-current light source should be used to avoid flickering.

The proposed visual measurement system should work in a real-time manner. EN 13848-2:2006 requires all measurements to be sampled at constant distance-based intervals not larger than 0.5 m [32]. The camera used in this study was set to sample at 30 FPS to ensure the high resolution of 1920 × 1080. That allows measurement for a vehicle running at 54 km/h. Therefore, a camera supporting a higher sample rate is desired for installation on in-service vehicles, considering that a regional train’s average speed is usually between 70 and 90 km/h. A higher sample rate will require processing hardware with higher processing speed or efficiency improvements in the data processing algorithm. It is worth noting that the traditional TG standards of the EN 13848 series are applicable for TG inspection, which is carried out at relatively large time intervals, such as every six months. Therefore, this standard series has strict requirements for inspection devices. The proposed visual measurement system was designed as a monitoring device installed on in-service vehicles, delivering a much larger amount of condition data and allowing statistical processing. This may compensate for some drawbacks of monitoring devices such as a low sample rate.

Weather variation may cause the performance degradation of the proposed system. In particular, in a harsh outdoor railway environment, variance in conditions is quite common, and the vehicles with the monitoring devices are deployed in different locations during different seasons. This raises high requirements for the robustness and the ability for easy adaptation of the algorithm. A detection and tracking algorithm based on a deep neural network may be a good candidate for meeting such requirements.

In conclusion, the developed system’s main advantage is its easy integration, low cost, and energy efficiency. The developed algorithm was validated and provided results comparable to those for a reference measurement system at lower speeds. This proves the suitability and applicability of the principles used in the proposed visual measurement system for wheel–rail lateral positioning, track gauge evaluation, and lateral alignment estimation after combining it with the inertial system output.

## Figures and Tables

**Figure 1 sensors-21-01297-f001:**
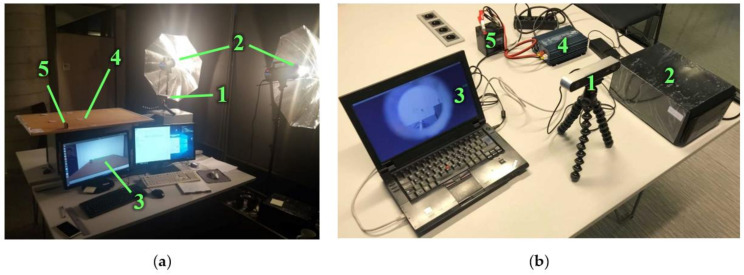
Test platform for real-time testing of sensing solutions: (**a**) laboratory test setup: 1—stereo camera with tripod; 2—light sources; 3—video output; 4—measurement area; 5—measured object; (**b**) measurement platform for field tests: 1—stereo camera with tripod; 2—NVIDIA Jetson TX2; 3—notebook computer; 4—12/220V inverter; 5—12-volt battery.

**Figure 2 sensors-21-01297-f002:**
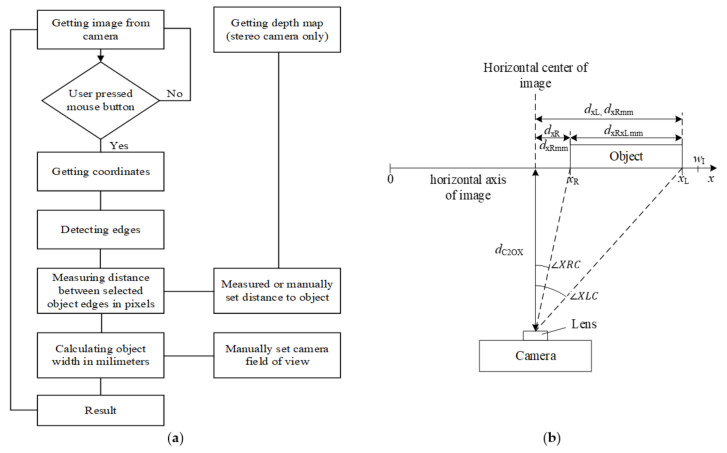
Object’s width measurement: (**a**) algorithm; (**b**) scheme.

**Figure 3 sensors-21-01297-f003:**
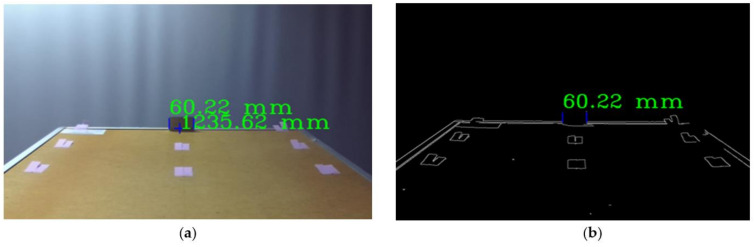
Example of images used in the experiment: (**a**) captured image if distance and width overlayed; (**b**) edge detection image used for automatic selection of vertical edges of the object at a specified point.

**Figure 4 sensors-21-01297-f004:**
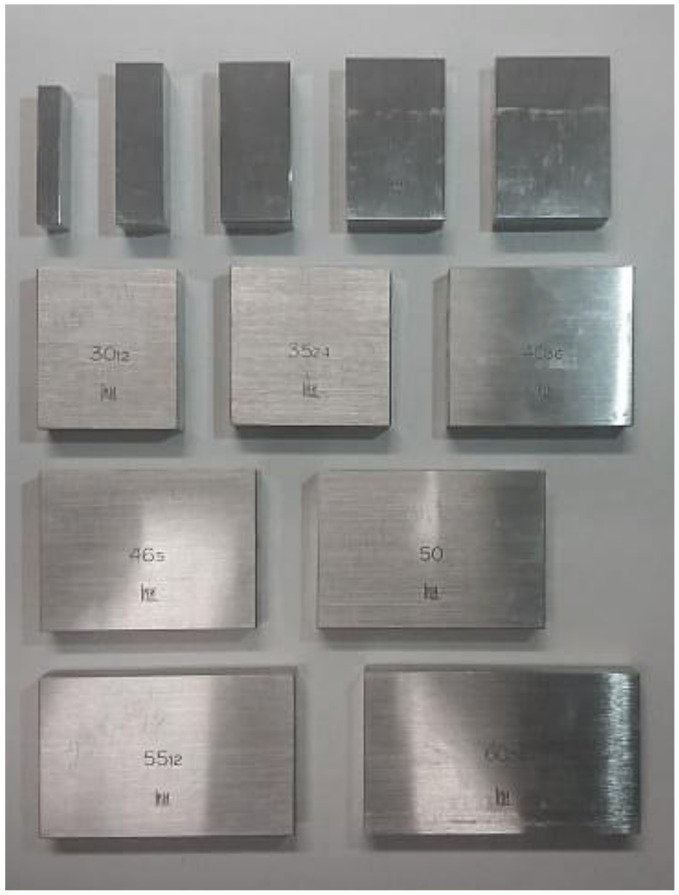
Specimens used during the experiment.

**Figure 5 sensors-21-01297-f005:**
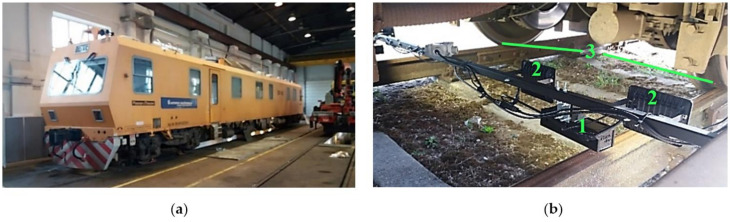
Measurement system installation: (**a**) railway geometry-measuring machine Plasser & Theurer EM-140; (**b**) mounted equipment: 1—stereo camera in a protective case; 2—artificial light sources; 3—measurement areas.

**Figure 6 sensors-21-01297-f006:**
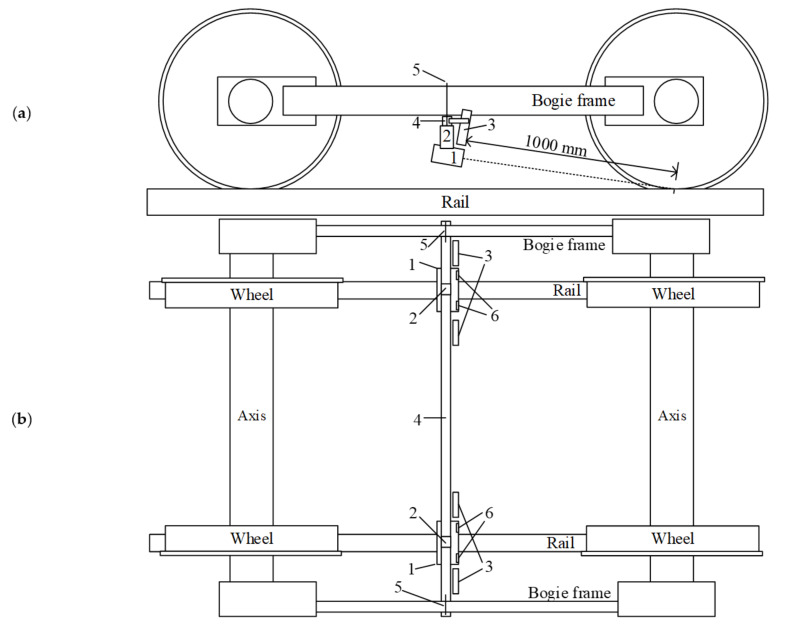
Schematic view of whole setup and camera position during field tests: (**a**) side view; (**b**) top view: 1—camera housing with Stereo Labs ZED stereo camera; 2—damper; 3—light source; 4—frame of the measurement system; 5—fixation; 6—camera lenses.

**Figure 7 sensors-21-01297-f007:**
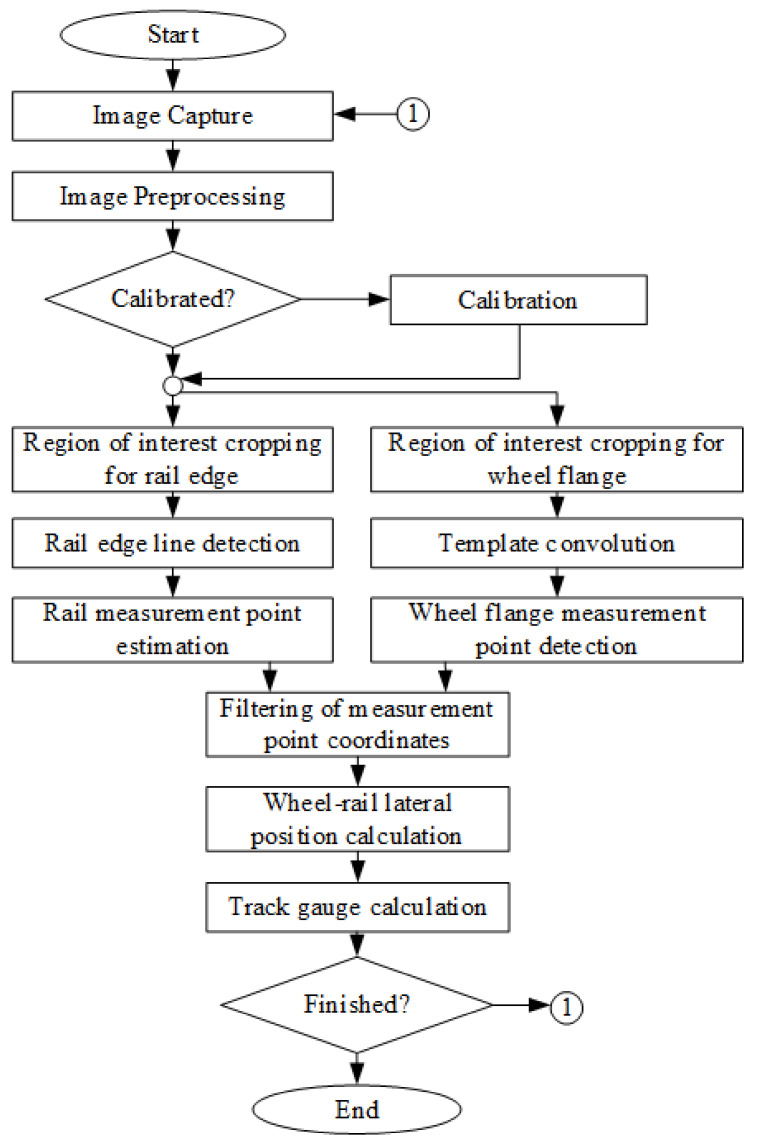
The wheel–rail lateral position measurement algorithm.

**Figure 8 sensors-21-01297-f008:**
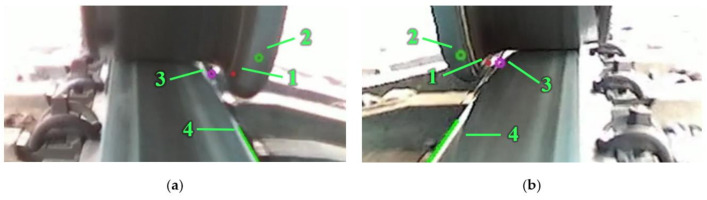
Real-time processing visualisation: (**a**) left wheel and rail; (**b**) right wheel and rail. In both: 1—detected wheel flange measurement point; 2—estimated rail edge point; 3—estimated wheel flange measurement point; 4—detected rail edge line.

**Figure 9 sensors-21-01297-f009:**
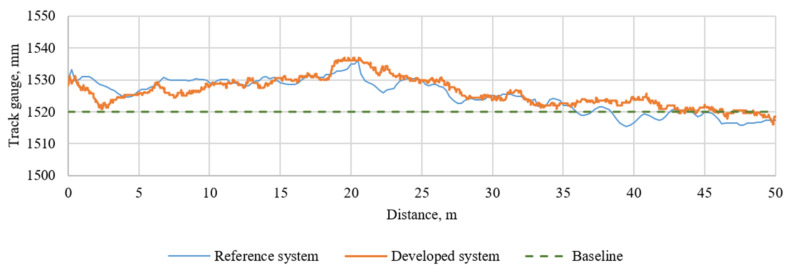
Comparison of rail gauge measurements.

**Table 1 sensors-21-01297-t001:** Experimental measurement results in millimetres.

Real value	5.12	10.24	15.36	21.50	25.00	30.12	35.24	40.36	46.5	50.00	55.12	60.24	65.36
Measured value	5.33	10.37	15.43	21.26	25.00	30.14	34.93	40.16	46.55	49.73	55.38	60.46	65.37
Error	−0.21	−0.13	−0.07	0.24	0.00	−0.02	0.31	0.20	−0.05	0.27	−0.26	−0.22	−0.01
Stand. dev.	0.33	0.32	0.35	0.36	0.28	0.28	0.35	0.39	0.32	0.43	0.35	0.40	0.38
Min. value	4.77	9.80	14.79	20.74	24.36	29.77	34.36	39.44	45.96	49.32	54.79	59.67	64.62
Max. value	5.70	10.86	15.95	21.74	25.74	30.79	35.73	40.74	47.19	50.35	55.8	61.05	66.16

## Data Availability

The data presented in this study are available on request from the corresponding author.

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
