# Peer review of "Visual Measurement System for Wheel–Rail Lateral Position Evaluation"

_sensors, 2021, doi:10.3390/s21041297_

Round 1

Reviewer 1 Report

Dear Authors,

Thank you for conducting a very interesting study to publish it in an international journal.

This publication is very interesting.

The structure of the work is very well suited to the goal. In many places of work you can see a deep involvement and knowledge of the authors' research topics.

The high value of the work consists in the efficient authors arrangement of the practical study, as well as consistent application of the created methodological workshop in well conducted own research. Great.

It is worth combining "Railway Engineering, Timetable Planning and Control, Artificial Intelligence and Externalities" with surveying and diagnostic monitoring of rail transport infrastructure. In such a way that it is mobile and does not restrict the timetable.

Surveying and diagnostic monitoring has a significant impact in railroad engineering, the impact on infrastructure maintenance systems and superstructure on the timetable and its control. Once again - interesting work.

Further information on various issues identified in the manuscript appears below:

  1. In the introduction: controversy and divergent hypotheses should be highlighted and indicate the main conclusions. Next, information on where the research was conducted, on what railroad, or directly in the field, make characteristics of the research object (whether the track gauge parameter was tested on straight track sections, in a curve, on a transition curve), EM-140, whether the facility was only under laboratory conditions.
  2. All abbreviations should be explained in the publication (for example: TSC, TCS, TGC, TRC, GNSS, IMU, LIDAR, EM-140 - Elektronischer Messtriebwagen, Elektronischer Messwagen).
  3. The sentence in lines 79-80 is not entirely truthful "It should be noted that the TG measure ment technique itself is not perfect, and new technical solutions are needed". It misleads the reader and academics; one gets a sense of inadequacy. Why do the authors claim this?

In surveying and diagnostics of rail transport in the field of track geometry monitoring there are two main methods: direct and indirect. Monitoring of track geometry depending on what category of line (the track is built in: main line, local significance) - is monitored cyclically in different time horizons. It covers a wide range of geometric parameters in the horizontal and vertical plane. Including in reference to surveying Railway Special Grid -  signs. So where does the main problem lie: "It should be noted that the TG measure ment technique itself is not perfect, and new technical solutions are needed."

  1. The paragraph contained in sentences 154 - 161 should be moved to Chapter One - (at the conclusion of this section).
  2. The same section title is shown twice:2) and 3) Materials & Methods

It should be corrected.

  1. Related work (Section 2) - it is recommended to divide in terms of three main themes:

- existing direct methods - tabular overview (of measurement systems and used sensors and measured track geometry parameters);

- existing indirect methods (of measurement systems and used sensors and measured track geometry parameters);

- connection to surveying Railway Special Grid, next sRSG TLS.

Then reference is made both to the EN standard but also to TSI - on the technical specifications for interoperability, regulations and national regulations concerning track geometry.

  1. Literature review is not rigorous. More literature on the safety and infrastructure railway (turnouts, railway line, surveying Railway Special Grid): for example, lots of recent work done by researchers. Literature and discussion should be supplemented.
  2. It would be helpful to correct the name of Figure 4.
  3. Sentence 205-207. Think about it, consider whether the rangefinder (laser distance meter - LDM) used - was sufficient in terms of accuracy.
  4. It is worth correcting the title of figure 6. Add a main title, then sub-titles for figures (a) and (b).
  5. All equations should be referenced in the text.
  6. Figure 9 - correct the name of the „track gauge” parameter.
  7. Sentence 405 – 410. If the authors have the results of the light value measurements - they should be indicated in the study, for laboratory conditions and for field conditions.

This completes the review.

Kind regards

Reviewer 2 Report

The investigation was intended to focus on developing and applying a low-cost sub-system that allows the detection of the wheelset-track lateral position to extend the functionality of existing inertial track geometry measurement systems.

This is an interesting, but not well organized and written paper. The main ideas are quite well developed, but some integration/adjustments should be done:

  • Do not use abbreviations in abstract
  • The abstract of the paper should be expended by giving more specific results
  • The use of non-technical words should be avoided.
  • The quality of the figures should be improved to a better understanding
  • What are the lessons learned from the conclusions other than the observation?
  • The novelty is of this article or is it a validation of a previously established notion/fact?

Reviewer 3 Report

Title: Visual measurement system for wheel-rail lateral position evaluation 

Adequate, because it corresponds to the content of the article.

Abstract: Suitable, clear and objective.

Keywords: appropriate and related to the content of the article. "Track gauge measurement" should be added.

Text: Clear and correct.

Itemization: Suitable, features appropriate content development.

Bibliographic research: very well developed and very focused on the scope of the article.

Problem studied: relevant

Suggested solution: creative, interesting and appropriate

  • Objective of the study: clear and properly defined.
  • Proposal for a new method of measuring track gauges and wheel axles.
  • The mathematical formulations used are justified.
  • Promising results.

Conclusions:

  • Consistent with the scope of the article.

References: in adequate numbers and related to the content of the article. Predominantly composed of qualified and recent articles. There is no reference to an article published by Sensors - Open Access Journal.

Additional questions:

In the measurements shown in table 1, is the camera's position angle similar to the camera's position in field measurements?

How was the immobilization of the camera guaranteed in field measurements?

Present a schematic drawing of the camera's position in field measurements.

Analyze the loss of accuracy of field measurements as a function of increased vibrations due to vertical irregularities.

Conclusion of the Review and Final Suggestion:

The article can be accepted after the answers to the doubts are incorporated into the text.

Round 2

Reviewer 2 Report

The paper is now accepted as is.